# Sp1 Upregulation Bolsters the Radioresistance of Glioblastoma Cells by Promoting Double Strand Breaks Repair

**DOI:** 10.3390/ijms241310658

**Published:** 2023-06-26

**Authors:** Xiongxiong Liu, Chao Sun, Qiqi Wang, Ping Li, Ting Zhao, Qiang Li

**Affiliations:** 1Institute of Modern Physics, Chinese Academy of Sciences, Lanzhou 730000, China; lxx002@impcas.ac.cn (X.L.); zhaoting@impcas.ac.cn (T.Z.); 2Key Laboratory of Heavy Ion Radiation Biology and Medicine, Chinese Academy of Sciences, Lanzhou 730000, China; 3Key Laboratory of Basic Research on Heavy Ion Radiation Application in Medicine, Lanzhou 730000, China; 4University of Chinese Academy of Sciences, Beijing 100049, China; 5College of Life Science, Northwest Normal University, Lanzhou 730030, China

**Keywords:** glioblastoma, Sp1, DNA-PKcs, radioresistance, DSB repair

## Abstract

Radioresistance remains a critical obstacle in the clinical management of glioblastoma (GBM) by radiotherapy. Therefore, it is necessary to explore the molecular mechanisms underlying radioresistance to improve patient response to radiotherapy and increase the treatment efficacy. The present study aimed to elucidate the role of specificity protein 1 (Sp1) in the radioresistance of GBM cells. Different human GBM cell lines and tumor-bearing mice were exposed to ionizing radiation (IR). Cell survival was determined by the colony formation assay. The expression of genes and proteins in the cells and tissues was analyzed by RT-PCR and western blotting, respectively. The γ-H2AX, p-Sp1 and dependent protein kinase catalytic subunit (DNA-PKcs phospho S2056) foci were analyzed by immunofluorescence. Apoptotic rates were measured by flow cytometry. Sp1 was upregulated after IR in vitro and in vivo and knocking down Sp1-sensitized GBM cells to IR. Sp1 activated the DNA-PKcs promoter and increased its expression and activity. Furthermore, the loss of Sp1 delayed double-strand breaks (DSB) repair and increased IR-induced apoptosis of GBM cells. Taken together, IR upregulates Sp1 expression in GBM cells, enhancing the activity of DNA-PKcs and promoting IR-induced DSB repair, thereby leading to increased radioresistance.

## 1. Introduction

Glioblastoma (GBM) is adults’ most common primary brain tumor [1]. Radiotherapy is the usual approach in a critical part of the clinical management of both newly diagnosed and recurrent GBM [2,3], although radioresistance severely limits the therapeutic outcomes. GBM recurrence is frequent even after high-dose radiation therapy [4], and no standardized clinical approach is available to sensitize GBM tumors to radiotherapy. Ionizing radiation (IR) promotes tumor cell death by inducing DNA double-strand breaks (DSB) and other forms of DNA damage [5]. However, activating the DSB repair pathways can protect the cells against apoptosis, leading to radioresistance. Therefore, it is crucial to fully understand the molecular mechanisms regulating DSB repair and identify targets for reversing the radioresistance in cancer cells.

Specificity protein 1 (Sp1) is the first identified transcription factor crucial in cancer development [6]. It is overexpressed in most cancers and is associated with poor clinical outcomes [7,8,9]. Sp1 is phosphorylated at Ser101 by ATM/ATR in response to DNA damage [10,11,12], and its phosphorylated form transcriptionally regulates the DNA-dependent protein kinase catalytic subunit (DNA-PKcs) in daunorubicin-resistant leukemia cell lines [13]. Our previous study revealed that DNA-PKcs is involved in DSB repair [14]. Furthermore, Hosoi et al. showed that DNA-dependent protein kinase (DNA-PK) activity and Sp1 are upregulated in colorectal cancer cells [15]. Thus, according to these findings, our hypothesis was that Sp1 might regulate we hypothesized that Sp1 regulates DSB repair through DNA-PKcs, although the specific mechanism remains unknown.

Therefore, the present study aimed to explore the role of Sp1 in the induction of radioresistance in GBM using a combination of bioinformatics and molecular biology.

## 2. Results

### 2.1. IR Upregulated Sp1 Expression in GBM Cells

The expression of Sp1 in GBM and normal brain tissues was analyzed, and its impact on patient prognosis was evaluated by bioinformatics. Sp1 gene expression was significantly up-regulated in the GBM tissues compared to that in normal tissues (*p* < 0.001; Figure 1A), and high Sp1 expression was correlated to poor survival (*p* = 0.043) and poor prognosis (AUC = 0.817, CI = 0.798–0.837) (Figure 1B,C). In addition, previous studies showed that IR induces changes in the expression levels of radioresistance-related molecules [16]. Consistent with this, our results showed a dose-dependent increase in the expression of Sp1 mRNA in all GBM cell lines (M059K, U87, T98G and U251) 24 h after the exposure to 0, 2, 4 and 6 Gy IR (Figure 1D). A similar trend was also observed for Sp1 protein expression (Figure 1E,F).

### 2.2. IR Upregulated Sp1 Expression in GBM Cells In Vivo

Subcutaneous U87 tumors were established in Balb/c nude mice, which were irradiated with 10 Gy single fraction X-ray to assess whether IR induced Sp1 upregulation in vivo (Figure 2A). The tumors were harvested on days 1, 3, 5, 7 and 9 post-irradiation. Sp1 mRNA was significantly upregulated in the tumors of the irradiated mice compared to its expression in the non-irradiated group (*p* < 0.01) (Figure 2B). Sp1 protein reached the highest expression on day five post-irradiation (Figure 2C,D). The immunohistochemical analysis of the tumor tissues also showed a remarkable increase in the expression of Sp1 protein on day five post-irradiation (Figure 2E). The results revealed that IR significantly upregulated the expression of Sp1 mRNA and protein in vitro and in vivo.

### 2.3. Sp1 Knockdown Radiosensitized GBM Cells In Vitro

To further determine whether Sp1 expression level correlates with radioresistance, Sp1 expression in the GBM cells was silenced using specific siRNA (Figure 3A,C,E,G) and analyzed their potential proliferative post-irradiation. and their proliferative potential after the irradiation was assessed to determine whether Sp1 expression was correlated with radioresistance. Sp1 knockdown significantly increased the radiosensitivity of all the GBM cell lines, and the sensitizer enhancement ratios at 10% survival level (SER10) were 1.34, 1.32, 1.42 and 1.41 for U87, T98G, M059K and U251 cells, respectively (Figure 3B,D,F,H).

### 2.4. Sp1 Knockdown Radiosensitized GBM Cells In Vivo

The effect of Sp1 on the radiosensitivity of GBM cells was also evaluated in vivo. The cells were transduced with lentiviruses expressing the Sp1-specific shRNA or the scrambled control (Figure 4A), and Sp1 knockdown was confirmed by western blotting and RT-PCR (*p* < 0.01; Figure 4B,C). The knockdown of Sp1 significantly increased the inhibitory effects of IR on tumor growth compared to Sp1 knockdown or IR alone (Figure 4D,E), which was consistent with the in vitro findings.

### 2.5. Sp1 Activated the DNA-PKcs Promoter and Enhanced DNA-PKcs Expression and Activity

The “Correlation Analysis” module of the GEPIA2 data platform was used to analyze the correlation between Sp1 and DNA-PKcs (parameters set as Gene: Sp1/DNA-PKcs; Correlation Coefficient: Pearson; TCGA Tumor: CESC Tumor). A significant positive correlation between Sp1 and DNA-PKcs was observed, and the correlation coefficient R was 0.65 (Figure 5A, *p* < 0.001). Our previous study demonstrated that DNA-PKcs is activated in response to DNA damage and is involved in the DSB repair pathway [14]. Therefore, a dual-luciferase reporter assay was performed to assess whether Sp1 mediates the activity of the DNA-PKcs promoter. Putative Sp1 binding sites within the promoter region of DNA-PKcs were evaluated using JASPAR (https://jaspar.genereg.net/ (accessed on 5 June 2020), and a binding site with high G + C content and lack of CCAAT and TATA-like elements at −78 bp to −55 bp was selected (Figure 5B). DNA fragments containing wild-type or mutated sites were cloned into the promoter region of a firefly luciferase reporter plasmid (pGL3), and then transfected into HEK 293Tcells. Figure 5C shows that the luciferase activity in the HEK 293T cells co-transfected with Sp1 overexpression plasmid (pCMV6-Sp1) and wild-type DNA-PKcs promoter plasmid (pGL3-p-WT) was significantly higher than that of the other groups, suggesting that the selected sites were involved in the transcription of DNA-PKcs. As expected, the luciferase activity was markedly reduced when the mutated (MUT) promoter was transfected (Figure 5C). Taken together, these results revealed that Sp1 activated the DNA-PKcs promoter.

The ectopic expression of Sp1 in U87 cells significantly upregulated DNA-PKcs (Figure 5D), while the silencing of Sp1 resulted in a modest inhibitory effect (Figure 5E), indicating a potential correlation between DNA-PKcs expression and the function of Sp1. The cells exposed to IR showed a similar trend in DNA-PKcs expression compared to the cells over-expressing Sp1, indicating that the upregulation of Sp1 can increase DNA-PKcs expression regardless of IR. Sp1 expression was also positively correlated to the activation of DNA-PKcs, i.e., p-DNA-PKcs (ser2056). Furthermore, the silencing of Sp1 and the subsequent irradiation did not increase the expression of DNA-PKcs nor the activity of DNA-PKcs (Figure 5E). Overall, the above results suggested that Sp1 was necessary for enhancing DNA-PKcs expression and activity.

### 2.6. p-Sp1 Was Located in the DSB Sites

After irradiation, sp1 phosphorylation and recruitment in response to DNA damage were analyzed by immunofluorescence at various time points (0.5, 1, 2, 12 and 24 h). Figure 6 shows that p-Sp1 (Ser101) and γ-H2AX expression kinetics were similar. The number of p-Sp1 and γ-H2AX foci peaked 1 h post-irradiation in U87 and M059K cells and decreased in a time-dependent manner. Moreover, p-Sp1 was co-localized with γ-H2AX in the irradiated cells, and the foci were on close proximity to DNA DSBs, thus underscoring the role of Sp1 in DSB repair in GBM cells.

### 2.7. Sp1 Silencing Delayed IR-Induced DSB Repair

The formation of γ-H2AX foci, an indicator of lethal DNA damage [17], was assessed in the irradiated Sp1-knockdown cells (Figure 7A,B). The number of γ-H2AX foci peaked 1 h post-irradiation in the negative control (NC) and the si-Sp1- U87 and M059K cell lines but decreased significantly in the NC group in a time-dependent manner. In contrast, the foci did not change till 24 h post-irradiation in the si-Sp1 group (Figure 7C–F). These findings indicated that Sp1 facilitated DNA damage repair, and the ablation of Sp1 expression delayed DSB repair.

### 2.8. Sp1 Silencing Aggravated IR-Induced Apoptosis In Vivo and In Vitro

Sp1 silencing markedly enhanced the apoptotic rates in U87 and M059K cells 48 h post-irradiation (Figure 8A,B). The apoptosis rates in the Sp1-knockdown U87 and M059K cells after irradiation were 49.57 ± 8.71% and 48.53 ± 9.93%, respectively (Table 1), significantly higher than those in the IR group. Consistent with this, numerous TUNEL-positive apoptotic cells were observed in the tumor tissues from the si-Sp1 + IR group (Figure 8C), and their number was approximately 35% higher than that in the NC + X-ray group (Figure 8D).

## 3. Discussion

Radioresistance is a major barrier to effective radiotherapy outcomes in GBM patients [18]. Activation of the DNA damage response pathway and the resulting increase in the DNA repair capacity mainly contribute to developing GBM radioresistance [19,20,21]. Therefore, it is crucial to understand its underlying mechanisms to improve the therapeutic outcomes. Sp1 regulates the expression of specific genes (i.e., MAPK, p38, JAK/STAT and PI3K/Akt), facilitating the post-translational modification of Sp1 and affecting its stability. Furthermore, the expression of the Sp1 protein is tightly regulated to maintain cellular homeostasis [7]. Grinstein E et al. [22] discovered that Sp1 protein expression is predominant during the G1 phase of the cell cycle, being an important regulator. The knockdown of Sp1 induces G1 phase arrest and suppresses cell proliferation, clonogenicity and the expressions of stem cell markers in nasopharyngeal carcinoma [23]. However, this study found that IR significantly upregulated Sp1 in multiple human GBM cell lines and xenograft-bearing mice. Luo J. et al. [24] found that Sp1 promotes the proliferation of glioma cells. This is likely due to the increased requirement of Sp1 for the activation of DSB repair genes, which leads to the upregulation of Sp1 through a feedback effect. Yu, Y. et al. [25] demonstrated that Sp1 downregulation reduces the radioresistance of U87 cells. The present study confirmed that silencing Sp1 in GBM cell lines markedly decreased their viability compared to the controls, indicating that IR-induced upregulation of Sp1 might be a reason for GBM radioresistance.

DNA damage and repair determine the fate of irradiated cells [26,27]. Sp1 transcriptionally activates multiple genes in response to DNA damage, including ATM, DDB1/2, NFBD1/MDC1 and XRCC1 [28,29]. One of the key proteins in DSB repair is DNA-dependent protein kinase (DNA-PK), which is composed of Ku (Ku70/80) and DNA-PKcs [14]. The inhibition of DNA-PK sensitizes GBM to radiation in mice [30]. As a core element of the DNA-PK, DNA-PKcs are essential in maintaining genomic stability [31]. Our previous report found that DNA-PKcs activation is essential for DSB repair [14]. Moreover, DNA-PKcs have a GC-rich promoter region. The present study revealed that the overexpression of Sp1 activated the DNA-PKcs promoter in U87 cells, which is consistent with another study showed that Sp1 enhances the activity of the DNA-PKcs promoter in daunorubicin-resistant leukemia cell lines (K562/DNR) [13]. The overexpression of Sp1 also upregulated DNA-PKcs, which led us to hypothesize that Sp1 lies upstream of DNA-PKcs. Sp1 overexpression also promoted the activation of DNA-PKcs, while the silencing of Sp1 significantly decreased the expression and activity of DNA-PKcs, resulting in delayed DSB repair and the sensitization of GBM cells to IR. Thus, based on these findings, our conclusion was that Sp1 mediated DSB repair by improving the activity of DNA-PKcs.

In addition, Sp1 phosphorylated at ser101 was recruited to the DSB sites and coincided with the disappearance of γH2AX, presumably indicating the completion of DSB repair (Figure 6). Furthermore, this study demonstrated that p-Sp1 and p-DNA-PKcs were co-localized at the DSB sites. However, the silencing of Sp1 markedly decreased p-DNA-PKcs expression and fluorescence intensity. Although knocking down DNA-PKcs did not affect Sp1 phosphorylation, it inhibited the formation of p-Sp1 foci, indicating that p-DNA-PKcs were recruited before p-Sp1 to the DSB sites. An interesting explanation could be that DNA-PKcs-mediated phosphorylation of Sp1 stabilized the NHEJ repair factors such as DNA-PKcs at the damage sites, thereby providing more spatiotemporal possibilities for DSB repair. IR induced Sp1 expression in GBM cells, promoting DNA-PKcs expression and activity. Since DNA-PKcs play a central role in DSB repair, inhibiting DNA-PKcs repressed NHEJ and delayed HR repair pathways [14].

Taken together, Sp1 upregulation contributed to the radioresistance of GBM cells by promoting DSB repair. The Sp1-DNA-PKcs pathway might be the major effector of DSB repair and radioresistance in GBM cells, raising the possibility of developing small-molecule inhibitors targeting Sp1 or DNA-PKcs, and potentially augmenting the efficacy of GBM therapy as radiosensitizers.

## 4. Materials and Methods

### 4.1. Cell Lines and Culture

Human GBM cell lines U251, U87 and T98G, and human embryonic kidney cell line HEK293T were obtained from the Cell Bank of the Chinese Academy of Sciences (Shanghai, China). The human GBM cell line M059K was obtained from the American Type Culture Collection. U251, U87, T98G and HEK293T cells were cultured in Dulbecco’s Modified Eagle Medium (DMEM) (Gibco, ThermoFisher Scientific, Shanghai, China) supplemented with 10% fetal bovine serum (FBS) (Minhai, Lanzhou, China). M059K cells were cultured in DMEM/F12 (Gibco-Invitrogen Corp, Grand Island, NY, USA) supplemented with 10% FBS (Corning, Mediatech, Manassas, VA, USA). All cells were maintained in a humidified incubator (Thermo Electro Corp, Waltham, MA, USA) at 37 °C under 5% CO_2_.

### 4.2. Irradiation

The cells and animals were irradiated using an X-ray generator (X-Rad 225, PXI Inc., Yuba City, CA, USA). Correction factors for attenuation were set, and the estimated dose rate was approximately 1.0 Gy/min.

### 4.3. RNA Extraction and RT-PCR

Total RNA was extracted using TRIzol reagent (Invitrogen, Carlsbad, CA, USA), and transformed into cDNA using the Applied Biosystems High-capacity cDNA Reverse Transcription Kit (QIAGEN, Frankfurt, Germany). Quantifast SYBR Green PCR kit (QIAGEN, Frankfurt, Germany) was used for RT-PCR, performed on the Quant Studio 5 Real-Time PCR System (Thermo Life tech ABI, Waltham, MA, USA). All assays were performed in triplicate, and the results were normalized to GAPDH. The primer sequences for Sp1 were as follows: F: 5′-TCACTCCATGGATGAAATGACA-3′, R: 5′-CAGAGGAGGAAGAGATGATCTG-3′.

### 4.4. siRNA Transfection and Establishment of Stable Sp1-Knockdown Cell Lines

The siRNA/shRNA sequences were designed and synthesized by Genepharma (Shanghai, China). The sequences of the specific siRNAs were as follows: DNA-PKcs 5′-AUAUUCUGAUUAAACCCUGTT-3′; Sp1 5′-GGAUGGUUCUGGUCAAAUATT-3′; negative control (NC) 5′-ACGUGACACGUUCGGAGAATT-3′. The cells were seeded in a 6-well plate, cultured till 50–60% confluence, and transfected with siRNA using Lipofectamine^®^ 3000 (Invitrogen, Carlsbad, CA, USA) according to the manufacturer’s protocol. The oligonucleotide sequence encoding short hairpin RNA (shRNA) (Sp1: 5′-GGATGGTTCTGGTCAAATA-3′) was cloned into the LV3 (H1/GFP & Puro) vector to construct the recombinant. The U87 cells were infected with the recombinant lentivirus, and the stably transduced cell lines (shSp1-U87) were obtained by puromycin screening.

### 4.5. Western Blotting

Proteins were extracted from the suitably treated cells using RIPA lysis buffer (Solarbio, Beijing, China) and separated by 10% SDS-PAGE as previously described [16]. The blots were probed with primary antibodies targeting the following proteins: p-Sp1 (Ser101) (1:1000, Active Motif, Carlsbad, CA, USA), DNA-PKcs (1:1000, ab32566, Abcam, Cambridge, UK), DNA-PKcs (phospho S2056) (1:1000, ab124918, Abcam, Cambridge, UK), γH2AX (Ser139) (1:1000, Cell Signaling Technology, Boston, MA, USA), Sp1 (1:3000, Proteintech Group, Wuhan, China) and GAPDH (1:10,000, Proteintech Group, Wuhan, China). Protein expression was quantified using the Quantity One software (Version 4.6.2, Bio-Rad, Hercules, CA, USA).

### 4.6. Immunofluorescence

Immunofluorescence assay was performed as previously described [32] using anti-γ-H2AX (S139) (1:500, Cell Signaling Technology, Boston, MA, USA) and anti-DNA-PKcs (S2056) (1:500, ab124918, Abcam, Cambridge, UK) antibodies. Phosphopeptide-specific antibodies were raised against serine-101 phosphopeptide, CTATQLpSQGAN and conjugated with KLH [33]. The nuclei were counterstained with Hoechst 33342 (Beyotime Biotechnology, Shanghai, China). The number of γ-H2AX, p-DNA-PKcs and p-Sp1 foci were counted in at least 50 cells per condition under a laser scanning confocal microscope (LSM 700, Zeiss, Oberkochen, Germany).

### 4.7. Dual Luciferase Assay

HEK293T cells were co-transfected with Sp1-overexpressing plasmid (OriGene, Rockville, MD, USA) or empty vector and the reporter plasmid for DNA-PKcs (PRKDC) promoter [Ppro-RB-Report-PRKDC-WT (−1000~−1)/Mut (−78~−55)] or corresponding empty plasmid (RiboBio, Guangzhou, China) using Lipofectamine^®^ 3000 (Thermo Invitrogen, Carlsbad, CA, USA). Dual luciferase assay was performed after 48 h using the Dual-Glo^®^ Reporter Assay System (Promega Biotech Co., Beijing, China) according to the manufacturer’s protocol. Luciferase activity was measured as the ratio of the firefly luciferase signal to the Renilla luciferase signal. All measurements were normalized to the control group, and the experiment was performed in triplicate.

### 4.8. Clonogenic Assay

Cell survival and proliferation rates were evaluated by the colony formation assay, as previously reported [34]. The colonies were counted under a stereomicroscope, and those with more than 50 cells were considered survivors.

### 4.9. Apoptosis Analysis

The apoptotic rates were quantified by flow cytometry as previously described using the Annexin V-FITC Apoptosis Detection Kit (BD, Franklin Lake, NJ, USA).

### 4.10. Xenograft Model

The studies involving animals were reviewed and approved by the Ethical Committee of the Institute of Modern Physics, Chinese Academy of Sciences. Five-week-old male Balb/c nude mice were obtained from Beijing HFK Bioscience (Beijing, China). Each mouse received a subcutaneous injection of 2 × 10^6^ U87 cells into one flank. The tumors were irradiated after 14 days (volume~0.3 cm^3^) with single-fraction 10 Gy X-rays. The mice were euthanized by cervical dislocation at 0, 1, 3, 5 and 7 days after irradiation. The tumors were immediately removed and frozen at −80 °C for subsequent experiments. Regarding immunohistochemistry, the stained slides were evaluated by whole slide imaging (Pannoramic MIDI and Viewer, 3DHISTECH). The tumor length and width were measured with a caliper, and the volume was calculated using 1/2ab^2^ (a: length, b: width).

### 4.11. Bioinformatics Analysis

The RNAseq data of 689 GBM (glioma) tissues and 1157 normal tissues were downloaded from TCGA and GTEX databases in TPM format from UCSC XENA (https://xenabrowser.net/datapages/ (accessed on 7 December 2021). The differential expression of Sp1 between the GBM and normal tissues was analyzed by the R-package “ggplot2”.

## Figures and Tables

**Figure 1 ijms-24-10658-f001:**
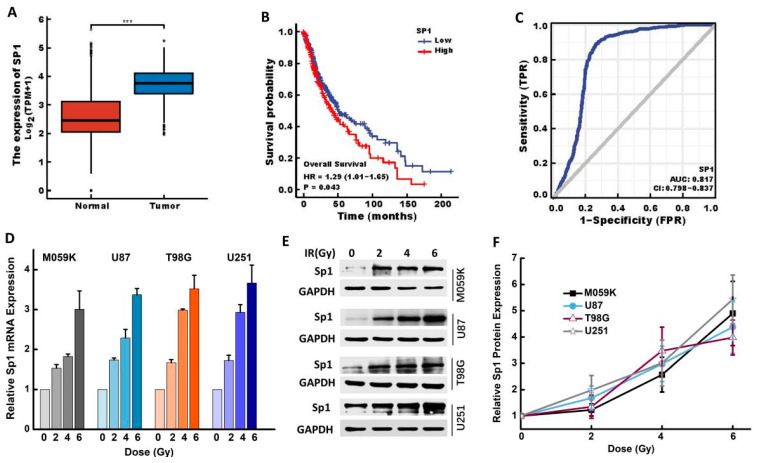
IR upregulated the Sp1 expression in GBM cells. Bioinformatics analysis results: (**A**) Sp1 gene differential expression between GBM and adjacent normal tissues (Mann-Whitney U test, *p* < 0.001). (**B**) Kaplan-Meier survival curves of GBM patients divided into Sp1-high and Sp1-low groups (log-rank test, *p* < 0.05). (**C**) Prognostic role of Sp1 in GBM. (**D**–**F**) Sp1 mRNA (**D**) and protein (**E**,**F**) expression in M059K, U87, T98G and U251 cells irradiated with 0, 2, 4 and 6 Gy X-ray. Results are expressed as mean ± SD for three independent experiments. *** *p* < 0.001 versus Normal.

**Figure 2 ijms-24-10658-f002:**
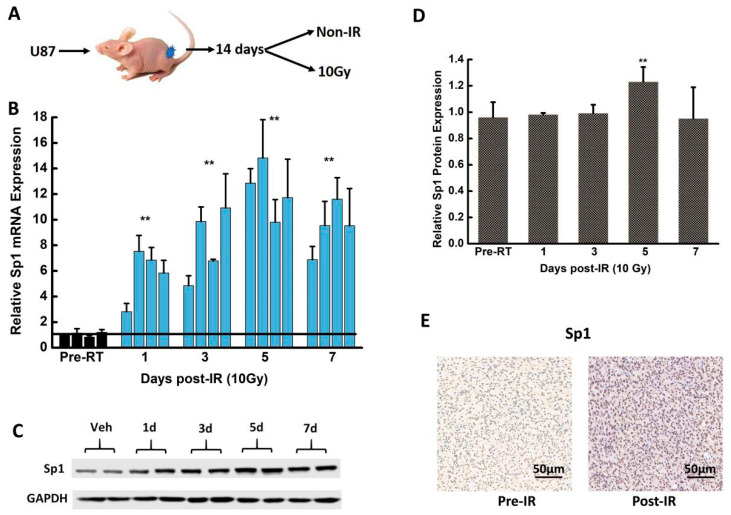
IR upregulated Sp1 expression in GBM cells in vivo. (**A**) Outline of the experimental procedure. (**B**,**C**) Sp1 mRNA (**B**) and protein (**C**) expression in U87-derived xenografts tumor from mice subjected to single-fraction 10 Gy X-ray. (**D**) Quantitative of Sp1 protein expression levels. (**E**) Representative images of U87-derived xenograft tumors harvested on day five post-irradiation showing the expression of Sp1 protein. ** *p* < 0.01 versus the Pre-RT group; n = 4 mice per group. Pre-RT: pre-radiotherapy.

**Figure 3 ijms-24-10658-f003:**
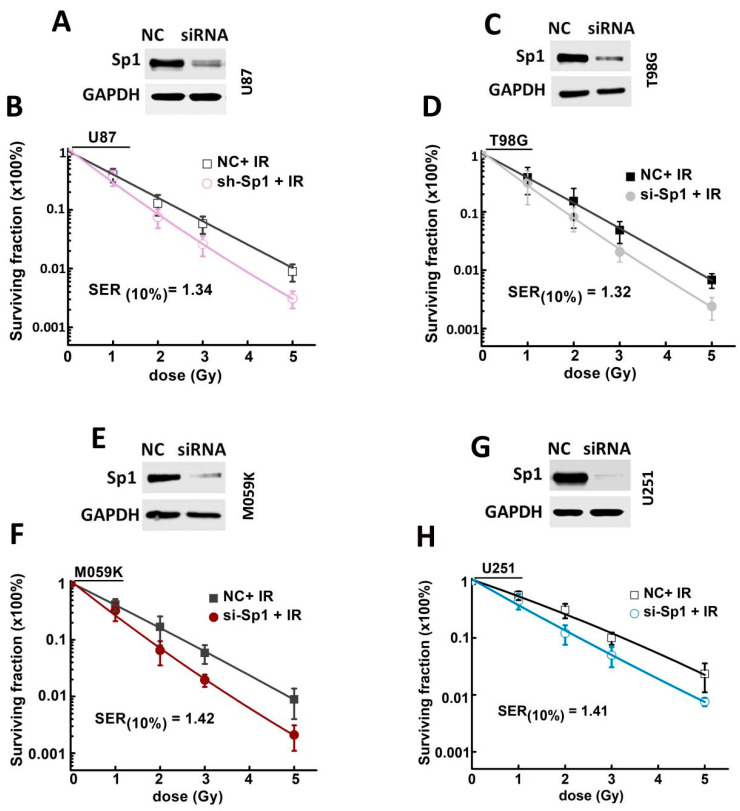
Sp1 knockdown radiosensitized GBM cells in vitro. Successful silencing of Sp1 by siRNA in (**A**) U87, (**C**) T98G, (**E**) M059K and (**G**) U251 cell lines. Clonogenic survival of (**B**) U87, (**D**) T98G, (**F**) M059K and (**H**) U251 cells irradiated with different doses of X-ray. Results are expressed as mean ± SD from three independent experiments.

**Figure 4 ijms-24-10658-f004:**
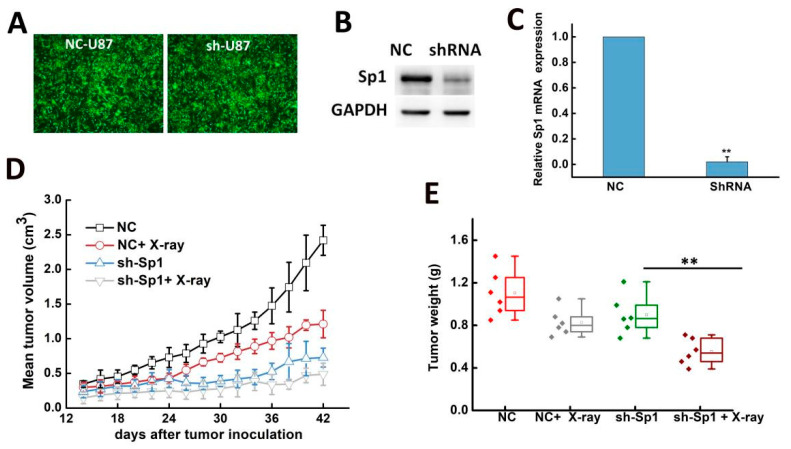
Sp1 knockdown radiosensitized GBM cells in vivo. (**A**) Representative images showing Sp1 protein expression (green fluorescence) in U87 cells after lentiviral transduction. The images were acquired using a fluorescence microscope (Olympus BX51, Tokyo, Japan) with 10× magnification. (**B**,**C**) Sp1 mRNA (**B**) and protein (**C**) expression in the indicated groups. ** *p* < 0.01 versus NC. (**D**) Mean tumor volume in the indicated groups was measured every other day from the 14th day after tumor cell inoculation. (**E**) Tumor weight in the indicated groups was measured on the 42nd day after tumor cell inoculation. ** *p* < 0.01 versus sh-Sp1 group; n = 4 mice per group.

**Figure 5 ijms-24-10658-f005:**
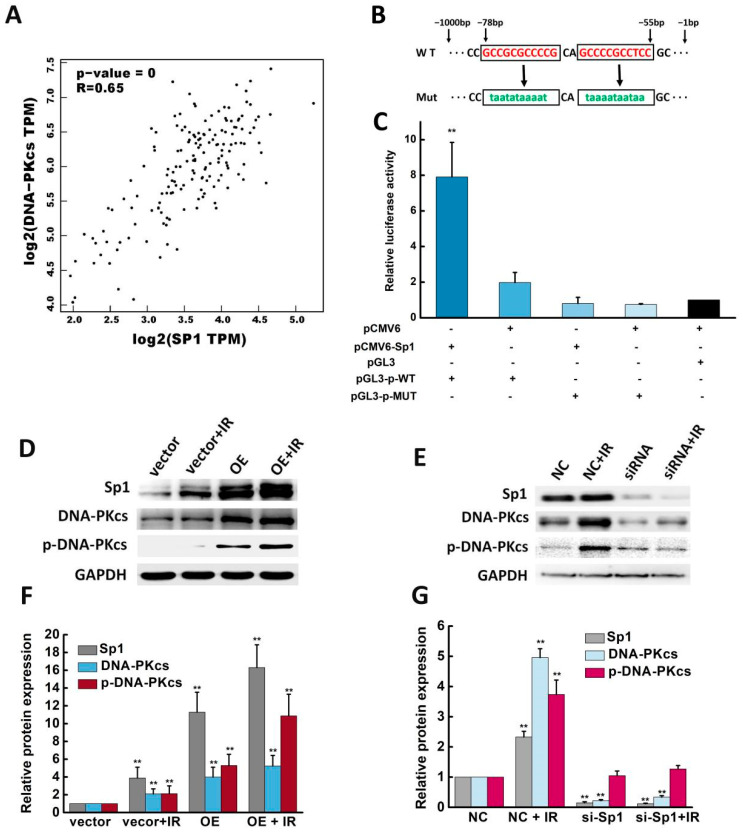
Sp1 activated the DNA-PKcs promoter and enhanced DNA-PKcs expression and activity. (**A**) The “Correlation Analysis” module of the GEPIA2 database was used to analyze the correlation between the Sp1 gene and DNA-PKcs (R = 0.65). (**B**) Schematic illustration of firefly luciferase reporter plasmids. (**C**) Relative luciferase activity in HEK 293T cells co-transfected with the indicated reporter constructs and Renilla luciferase plasmid. pCMV6 was used for Sp1 over-expression (OE), and pGL3 was the reporter plasmid containing the DNA-PKcs promoter. (**D**) Sp1 over-expression upregulated the expression and enhanced the activity of DNA-PKcs after irradiation. (**E**) Sp1 silencing downregulated DNA-PKcs expression and activity. (**F**,**G**) Expression of Sp1 and DNA-PKcs proteins in the indicated groups. ** *p* < 0.01 versus vector or NC group.

**Figure 6 ijms-24-10658-f006:**
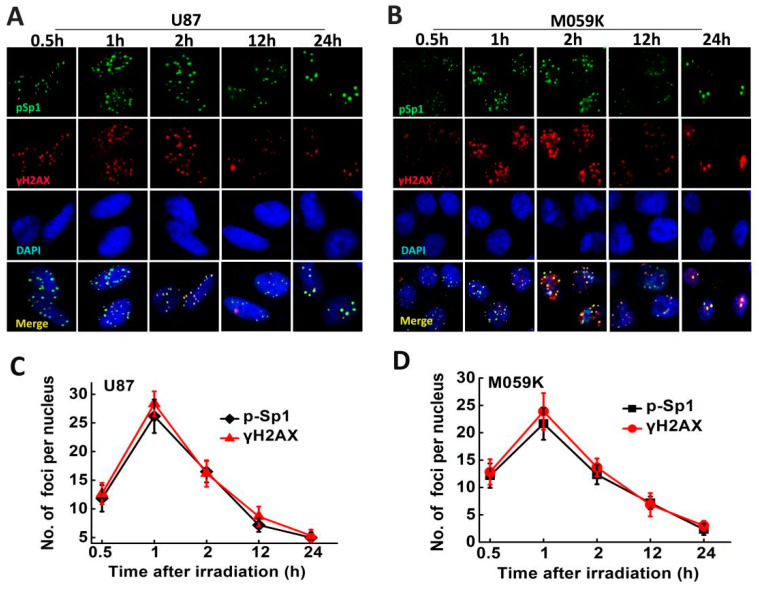
Phosphorylated Sp1 is located in the DSB sites after irradiation. (**A**,**B**) Representative images showing γ-H2AX and p-Sp1 foci in U87 and M059K cells 0.5, 1, 12, 24 h post-irradiation. The images were acquired using a Zeiss LSM-700 confocal microscope (Carl Zeiss, Jena, Germany) with 63× magnification. (**C**,**D**) Number of γ-H2AX and p-Sp1 foci in the indicated groups. Results are presented as mean ± SD from three independent experiments.

**Figure 7 ijms-24-10658-f007:**
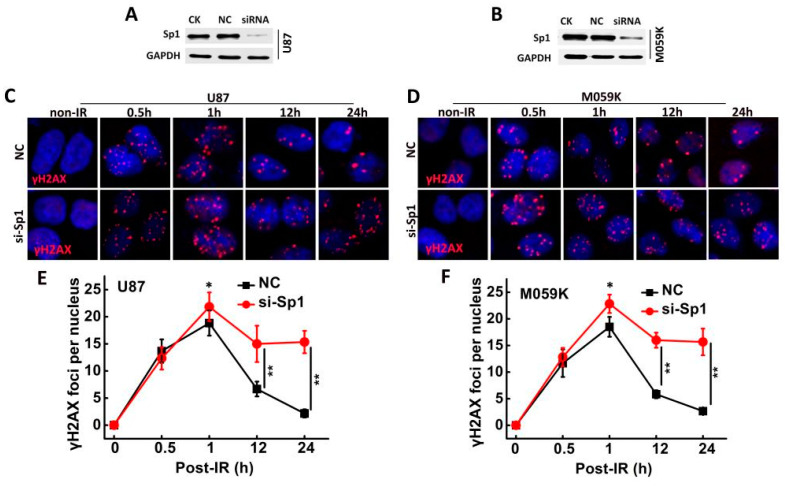
Sp1 silencing delayed IR-induced DSB repairs. Sp1 expression in (**A**) U87 and (**B**) M059K cells transfected with Sp1 siRNA. Representative images of γ-H2AX foci in (**C**) U87 and (**D**) M059K cell lines at various time points (0.5, 1, 12, 24 h) post-irradiation. The images were acquired using a Zeiss LSM-700 confocal microscope (Carl Zeiss, Jena, Germany) with 63× magnification. Number of γ-H2AX foci in (**E**) U87 and (**F**) M059K cells. Results are presented as mean ± SD from three independent experiments. CK: control, NC: transfected with negative siRNA. * *p* < 0.05, ** *p* < 0.01 versus the NC group.

**Figure 8 ijms-24-10658-f008:**
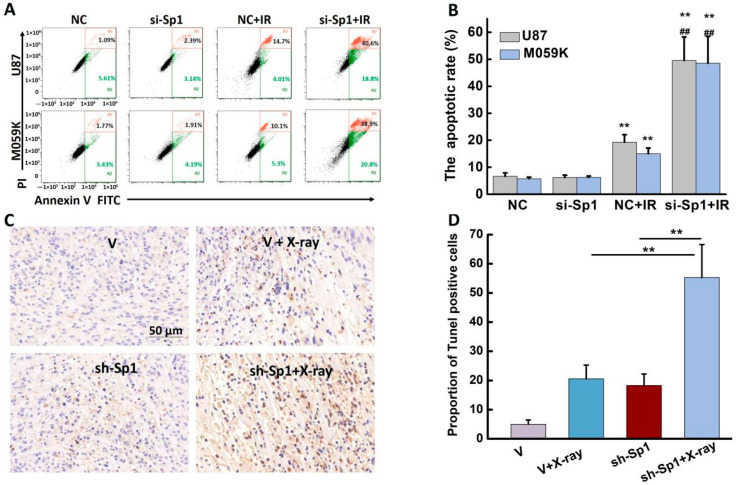
Silencing Sp1 promoted DSB repair and increased IR-induced apoptosis. (**A**) Representative images showing apoptotic U87 and M059K cells 48 h post-irradiation. (**B**) Apoptotic rates in the indicated groups. ** *p* < 0.01 versus NC group, ## *p* < 0.01 versus NC + IR group. NC: transfected with negative siRNA. (**C**) Representative images of TUNEL-stained tumor tissue sections from the indicated groups. (**D**) The proportion of TUNEL-positive cells. V: empty vector. Results are expressed as mean ± SD from three independent experiments.

**Table 1 ijms-24-10658-t001:** Effect of Sp1 silencing on the apoptotic rate of M059K and U87 cell lines at 48 h after IR.

Total Apoptotic Cells (%)		
	U87	M059K
NC	6.67 ± 1.26	5.76 ± 0.56
si-SP1	6.20 ± 0.87	6.26 ± 0.48
NC + IR	19.72 ± 2.79	15.03 ± 2.07
si-Sp1 + IR	49.57 ± 8.71 ***^###^	48.53 ± 9.93 ***^###^

*** *p* < 0.001 versus NC; ### *p* < 0.001 versus NC + IR. Results were obtained from three independent experiments.

## Data Availability

The data presented in this study are available in this article.

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
