# Peer review of "Sp1 Upregulation Bolsters the Radioresistance of Glioblastoma Cells by Promoting Double Strand Breaks Repair"

_ijms, 2023, doi:10.3390/ijms241310658_

Round 1
Reviewer 1 Report
The authors investigated the molecular mechanism behind radioresistance in glioblastoma using both in vivo, in vitro and bioinformatic models. They identify Sp1 as a major regulator of radioresistance through the promotion of DSB repair following irradiation.
From a quick literature search, Sp1 has been previously and frequently associated with cell proliferation, clonogenicity, apoptosis etc. Many papers have been published (10.1038/sj.onc.1205211, 10.1186/s12967-014-0222-1,https://doi.org/10.1091/mbc.E14-10-1443) to show that Sp1 downregulation can actually stop cell cycle, specifically at G1 phase. Therefore, a major concern is whether the effect we are seeing is due to an arrest of the cell cycle rather than any of the mechanisms explained by the authors. if the cells are arrested prior to any irradiation damage then the hypothesis of the authors would not be demonstrated. Things get more complicated when the authors state 'The cells exposed to IR showed a similar trend in DNA-PKcs levels compared to the Sp1-overexpressioning cells, indicating that upregulation of Sp1 can increase DNA-PKcs expression regardless of IR' implying that this mechanism may be indeed completely indipendent from irradiation/radioresistance. However, they conclude 'Overall, the data indicated that Sp1 increased DNA-PKcs expression level and activity in response to radiation exposure' which contradicts the data shown. At the very least the authors should show an analysis of the cell cycle phases distribution by flow cytometry
Overall, the authors will need to provide more explanations and evidence and at the very least discuss the limitations of this study and other potential interpretation of the data in the discussion. An extensive revision and change of scope is needed
More general comments:
Has a genomic profile of all cell lines been provided? This is extremely important to be provided to ensure that the cell lines are exactly what they are supposed to be
For the RT-PCR, it is advisable to use more than one normalizer and not just GAPDH as this has a very high variability in expression.
Remove 'title' from the title?
The protein increases observed in vivo are less convincing than those shown in the cell lines. The GAPDH increases too. I suggest the authors provide a quantification for the protein expression.
English must be improved
Author Response
Please see the attachment, thank you.

Reviewer 2 Report
1. Why did the authors use HEK293T cells for their promoter studies instead of GBM cells (Fig.5C)? The authors should also state in the text (and not only in the figure legend) that they used HEK293T cells for this study. The authors conclude based on the promoter study in HEK293T cells that "Taken together, Sp1 activates the DNA-PKcs promoter in the irradiated GBM cells (line 133)." This conclusion is wrong since the promoter study wasn't performed in GBM cells and the cells in the promoter study were not irradiated.
2. How do the authors explain that "Intriguingly, the combination of Sp1 silencing and IR decreased the activity of DNA-PKcs compared to IR alone (Fig. 5E), whereas the opposite was seen in the Sp1-overexpressing cells (Fig. 5D)"? Is this effect restricted to U87 or did they see the same effect in another cell line? How do they explain this effect?
3. The authors should mention that a recent study also shows that SP1 downregulation reduced the radioresistance of U87 cells (SP1 transcriptionally activates NLRP6 inflammasome and induces immune evasion and radioresistance in glioma cells - https://doi.org/10.1016/j.intimp.2021.107858).
4. The authors should mention that another recent study shows that inhibiting DNA-PK induces glioma stem cell differentiation and sensitizes glioblastoma to radiation in mice
minor points:
- The title should read SP1 upregulation bolsters....
- line 220 and 222: replace "30" and "14" with [30], [14]
Round 2
Reviewer 1 Report
No further comments